# Cellulase Immobilization on Nanostructured Supports for Biomass Waste Processing

**DOI:** 10.3390/nano12213796

**Published:** 2022-10-27

**Authors:** Aleksandrina M. Sulman, Valentina G. Matveeva, Lyudmila M. Bronstein

**Affiliations:** 1Department of Biotechnology, Chemistry and Standardization, Tver State Technical University, 22 A. Nikitina St., 170026 Tver, Russia; 2Regional Technological Centre, Tver State University, Zhelyabova St., 33, 170100 Tver, Russia; 3Department of Chemistry, Indiana University, 800 E. Kirkwood Av., Bloomington, IN 47405, USA; 4Department of Physics, Faculty of Science, King Abdulaziz University, P.O. Box 80303, Jeddah 21589, Saudi Arabia

**Keywords:** cellulase, immobilization, nanostructured supports, biomass

## Abstract

Nanobiocatalysts, i.e., enzymes immobilized on nanostructured supports, received considerable attention because they are potential remedies to overcome shortcomings of traditional biocatalysts, such as low efficiency of mass transfer, instability during catalytic reactions, and possible deactivation. In this short review, we will analyze major aspects of immobilization of cellulase—an enzyme for cellulosic biomass waste processing—on nanostructured supports. Such supports provide high surface areas, increased enzyme loading, and a beneficial environment to enhance cellulase performance and its stability, leading to nanobiocatalysts for obtaining biofuels and value-added chemicals. Here, we will discuss such nanostructured supports as carbon nanotubes, polymer nanoparticles (NPs), nanohydrogels, nanofibers, silica NPs, hierarchical porous materials, magnetic NPs and their nanohybrids, based on publications of the last five years. The use of magnetic NPs is especially favorable due to easy separation and the nanobiocatalyst recovery for a repeated use. This review will discuss methods for cellulase immobilization, morphology of nanostructured supports, multienzyme systems as well as factors influencing the enzyme activity to achieve the highest conversion of cellulosic biowaste into fermentable sugars. We believe this review will allow for an enhanced understanding of such nanobiocatalysts and processes, allowing for the best solutions to major problems of sustainable biorefinery.

## 1. Introduction

Scarcity of conventional fuels due to socioeconomic factors and reluctance of the communities to use them due to environmental aspects of their processing and applications increased the interest of industry in biomass or biomass waste processing to biofuels and value-added chemicals. There are many processing stages to convert lignocellulosic biomass to a direct source of biofuel, but the most important step is the decomposition of cellulose to fermentable (intermediate) sugars, which can be a feasible substrate for biofuel [1]. Environmentally favorable avenue for biomass processing is the use of enzymes which decompose cellulose to glucose [2,3,4,5]. However, low thermal and storage stability of enzymes as well as the presence of impurities, enzyme leakage, and a reusability problem are major shortcomings of employing free enzymes. These shortcomings can be minimized or even eliminated by immobilization of enzymes on various supports [6,7]. The role of support materials is to preserve the enzyme secondary structure as well as to create the favorable interactions with the enzyme [8]. The choice of a suitable carrier is also determined by the enzyme and process types [9,10].

In the last decade enzymes immobilized on nanostructured supports called *nanobiocatalysts* received considerable attention. Nanostructured supports are materials containing nanometer size features (normally, between 1 and 100 nm), such as nanoparticles (NPs) of different sizes and shapes including nanorods and nanofibers, materials with pores in a nanometer range, stimuli responsive nano-carriers, etc. [11,12,13,14,15,16]. This growing interest is explained by a possibility of nanobiocatalysts to overcome deficiencies of enzymes immobilized on traditional supports. Nanostructured supports minimize diffusion, thus significantly improving mass transfer. Additionally, these nanomaterials possess a high surface area for the enzyme immobilization, increasing the enzyme loading and improving their positioning on the surface. The latter often results in higher enzymatic activity [17,18].

In this review, we will discuss major features of immobilization of cellulase on nanostructured supports. Cellulases are enzymes that degrade cellulose—the most abundant natural polymer which forms plant cell walls. Cellulases are a cocktail of three enzymes: endoglucanases, exoglucanases, and *β*-glucosidases which are utilized to degrade different chemical bonds in cellulose [19]. For simplicity, here we will use the term “cellulase” (singular) which means the above cocktail, unless it is specified otherwise. It is worth noting that lignocellulosic biomass contains cellulose and hemicellulose (polysaccharides) as well as lignin (an aromatic polymer) [20]. The presence of lignin often inhibits the cellulose hydrolysis so special efforts are undertaken to overcome this problem. Besides stabilization and the possibility of catalyst reuse, the cellulase immobilization on nanostructured supports may reduce the cellulase surface charge, thus diminishing its non-specific binding to lignin and increasing the interactions with cellulose [21]. Often biomass waste first requires delignification with another enzymatic catalyst before cellulase can efficiently hydrolyze cellulose [22] or co-immobilization of several enzymes on the same support is implemented, which is a more prominent trend [23,24,25,26,27,28].

Because of the growing number of publications on cellulase immobilized on nanostructured supports, in this review we will discuss the above developments using the literature mainly published from the beginning of 2017 through August of 2022. It is important to mention preceding recent reviews on the topics related to cellulase immobilization on nanostructured supports [9,25,29,30,31,32,33,34,35,36,37,38,39,40,41,42,43] as they created the groundwork for our analysis.

We are also focusing on a combination of several crucial factors determining the performance of nanobiocatalysts, such as methods of cellulase immobilization, types of nanostructured supports, multienzyme nanobiocatalysts, etc. The structure of the review is presented in Figure 1.

## 2. Methods of Cellulase Immobilization

Typical methods of cellulase immobilization on nanostructured supports are analogous to those employed for other enzymes on various supports. They include adsorption on the support surface, encapsulation into the support, a covalent attachment, and cross-linking. In recent years, the covalent attachment became prevalent as it provides higher stability of immobilization and does not affect the enzyme structure if a tether used allows for protection of the enzyme secondary structure. On the other hand, encapsulation and cross-linking can be also beneficial for the biocatalyst performance if the enzyme conformation is preserved. Below we will discuss the above methods in more details using recent examples.

### 2.1. Adsorption

The major advantage of physical adsorption is its simplicity. It was used for immobilization of cellulase on metal-organic frameworks (MOFs) with high porosity [44,45], Fe_3_O_4_/acid activated montmorillonite composites of different structure [46], multiwall carbon nanotubes (MWCNTs) [47], etc. Thus, a high surface area is an important factor for successful adsorption as well as an opposite total charge of cellulase and the support, favorable pore size of the support to accommodate enzymes, etc. [48].

A pretreatment with ionic liquids (ILs) is known to facilitate hydrolysis of lignocellulosic biomass however ILs can degrade the enzyme [49,50]. Moreover, IL tolerance is an important parameter for in situ enzymatic saccharification of biomass to bioethanol in the presence of ILs, a promising industrial endeavor. Zhou et al. [44] studied IL tolerance of nanobiocatalysts based on several MOFs (with different metals) and physically adsorbed cellulase. Among four MOFs studied, ZIF-8 (MOFs which consist of Zn^2+^ and 2-methylimidazole ligands [51]) showed the highest enzyme adsorption capacity and remarkable tolerance to ILs. However, the most successful avenue for the protection of immobilized cellulase from the negative influence of ILs or desorption was found to be the modification of the support surface before or after cellulase adsorption. In this manner, a surface treatment of ZIF-8 with charge modifying compounds (chitosan) or hydrophobicity altering macromolecules (poly(ethylene glycol), PEG) allows for an increase of the enzyme loading capacity (Figure 1) [45].

The other authors demonstrated that modification of MOFs (UiO-66 which consists of zirconium clusters connected by 1,4-benzodicarboxylic acid [52]) with amino groups increases cellulase physical adsorption due to additional anchors [53]. An interesting modification of MOFs constructed around *Clostridium tyrobutyricum* Δ*ack*::*cat1*, with deleted *ack* gene and overexpressed *cat1* gene with further immobilization of cellulase allowed for the production of butyric acid via simultaneous saccharification and fermentation using rice straw as substrate [54]. Here, MOFs served both as ex-skeleton and photocatalyst to stimulate butyric acid production.

Post-enzyme-adsorption modification was demonstrated in a few papers. Cellulase adsorbed on MWCNTs and then protected by sodium alginate allowed for the increased nanobiocatalyst stability [47]. The gradual decrease of activity with each cycle was ascribed to weak, non-covalent bonds between cellulase and the support. An original method of cellulase immobilization was carried out by Zhu et al. who adsorbed the enzyme on Fe_3_O_4_@C NPs due to electrostatic interactions and then coated these NPs with a thin layer of sedimented silica, which enhanced cellulase adsorption, but still preserved its function [55]. Thus, the support modification/functionalization often allows for efficient nanobiocatalysts obtained by the cellulase adsorption. In the case of post-adsorption modification, the deposited outer layer should be sufficiently porous or swollen to allow cellulase a contact with the biomass waste source. Possible disadvantages of this method are the low loading and the loss of the enzyme, leading to the contamination of the final product.

### 2.2. Encapsulation

Encapsulation of cellulase can be carried out in three major ways. In one approach, cellulase molecules are trapped in the pores of porous materials, thus, the pore sizes play an important role in encapsulation. In the second approach, cellulase is encapsulated when the porous nanomaterial is formed. In the third approach, the enzyme is encapsulated by polymers, often during co-precipitation.

Zr-containing MOFs (UIO-66) with different mesopores (6.46, 7.55, 10.80 nm) were utilized to encapsulate several enzymes, including cellulase [56]. It is worth noting that cellulase has an ellipsoidal shape with a diameter of 4–6.5 nm and a length of 18–21.5 nm, therefore pores of suitable sizes can encapsulate the enzyme, although most probably adsorption also contributes in the immobilization process [57]. Surprisingly, the smallest pores allowed for the highest loading without jeopardizing the enzyme structure, as was verified by the comparison of the activity of free and immobilized enzymes. The specific pore structure of these MOFs (the presence of mesopores along with micropores) did not affect the mass transfer and allowed for an enrichment effect of the substrate, probably due to positioning of the latter at the support. An addition of 4.6 nm mesopores in microporous Zr-MOF obtained by biomineralization with dextran as sacrificial template allows better entrapment of cellulase within the material, improving loading capacity and stability of immobilized cellulase [58]. Mesoporous Zn-based MOFs were also utilized for cellulase encapsulation by simultaneous precipitation of the MOF precursors and cellulase (Figure 2) [59]. This significantly enhanced the cellulase loading and created structural defects during MOF formation (large pores), assisting in mass transfer and increasing enzymic activity.

Formation of nanogel in the presence of cellulase by direct cross-linking of poly(*N*-vinylpyrrolidone-*co*-*N*-methacryloxysuccinimide) with an enhanced green fluorescent protein allows successful encapsulation of the enzyme [60]. Self-assembly of chitosan around cellulase by salting out from a mixed solution allowed for the formation of nanohybrid which was deposited on alginate beads [49]. The nanobiocatalyst showed increased stability and efficiency in hydrolysis of sugarcane bagasse.

The most important advantage of encapsulation is robustness, although possible shortcomings, such as accompanied adsorption and a loss of conformation integrity could minimize its appeal. In addition, similar to the physical adsorption, cellulase encapsulation is only favorable, if the access to the enzyme in the nanomaterial is not impeded.

### 2.3. Covalent Attachment

Covalent attachment is frequently favored for cellulase immobilization because it provides enhanced stability which is often combined with improved activity of the enzyme—important advantages of this approach. Covalent immobilization, however, requires functionalization of the support unless the support inherently possesses functional groups [50,61,62,63]. In addition, a suitable linker is needed to preserve the enzyme conformation [9,64]. The most commonly used bifunctional linker is glutaraldehyde which interacts with amino groups at ambient conditions and does not require any catalyst [9,24,64,65]. Despite the length of glutaraldehyde is only 0.75 nm [66], it apparently provides a sufficient distance to prevent non-specific adsorption of the enzyme. Longer linkers, such as tetradecanedioic and docosanedioic dicarboxylic acids with approximate extended chain lengths of 1.4 and 2.2 nm, respectively, were also explored [67], however, the interaction of carboxyl group terminal linkers with amino functionalized supports (formation of a peptide bond) is less than favorable, requiring elevated temperatures or/and a catalyst [68,69].

In the case of supports with carboxyl groups on the surface (for example graphene oxide, GO), first acids are activated with carbodiimide (for example, 1-ethyl-3-(3-dimethylaminopropyl) carbodiimide) followed by the interaction with N-hydroxysuccinimide, thus, creating a functional group for attaching the enzyme [70].

Below we will discuss a few examples of the covalent attachment on various supports from the recent literature. Cellulase covalently immobilized on amino functionalized Fe_3_O_4_@SiO_2_ core-shell NPs provided high stability at various pH and temperatures in enzymatic saccharification of poplar wood [71]. This biocatalyst allows an enzymatic saccharification rate of 38.4% at 72 h, showing promise for deconstruction of lignocellulosic biomass. The same principle of cellulase immobilization using amino groups was utilized on a very different support: a hybrid conductive nanohydrogel prepared by polyaniline (PANI) nanorods formed on an electrospun cationic poly(ε-caprolactone) hydrogel containing cationic phosphine oxide macromolecule [72]. The hybrid nanobiocatalyst showed good performance in hydrolysis of cellulosic materials, exhibiting no loss of activity compared to free enzyme (Figure 3).

A proper functionalization of the support can be crucial for the efficient covalent enzyme attachment. This avenue was explored by Gao et al. who modified GO sheets using etherification with *p*-*β*-sulfuric acid ester ethyl sulfone aniline which creates a hydrophobic linker for further fast cellulase immobilization (~10 min) after diazotization [73]. It is noteworthy, however, that robustness of the fast attachment of cellulase is countered by a complex functionalization procedure, making it a questionable achievement.

In an original work, a sortase-mediated enzyme immobilization method (called sortagging) on microgels was developed and tested for five different enzymes including cellulase [74]. This method allows for a site-specific enzyme immobilization due to the covalent attachment on stimuli responsive microgel particles based on poly(*N*-vinylcaprolactam)/glycidyl methacrylate.

Thus, the possible disadvantage of the covalent attachment of cellulase is a complexity of chemical modification of the support and/or the enzyme.

### 2.4. Cross-Linking

Cross-linking of enzymes into aggregates leads to enzyme immobilization without the use of support materials, making it a robust approach. Here, the low density of aggregates is a crucial factor to allow a contact between cellulase and cellulosic biomass. Such cross-linking can be accomplished by a simple interaction with glutaraldehyde [75] or by more sophisticated methods. Activity of crosslinked enzyme aggregates (CLEA) was found to depend on the precipitant type, which could influence the CLEA density [76,77]. When cross-linked cellulase aggregates prepared by precipitation are combined with magnetic NPs, an extra advantage of easy magnetic manipulation of the nanobiocatalyst is added [78].

An original method of preparation of well-defined multienzyme hybrid nanoflowers (ECG-NFs) was proposed by Han et al. by cross-linking all three cellulase enzymes (cellobiohydrolase (CBH), endo-glucanase (EG), and *β*-glucosidase (BG)) and combining a binary tag consisting of elastin-like polypeptide (ELP) and *His*-tag [79]. Here, recombinant enzymes (EG-Linker-ELP-His (EGLEH), CBH-Linker-ELP-His (CBHLEH), and Glu-Linker-ELP-His (GLEH) (Figure 4), were assembled by incorporating a dual sticker (ELP-His) into the above enzymes. The nanoflowers formed catalyzed the cellulose hydrolysis into glucose with high pH, thermal, and storage stability as well as better catalytic activity compared to free enzymes. In this case, the open structure of enzyme aggregates is the key for the successful catalysis.

The possible shortcoming of this approach includes lower activity due to poor access to active site if cross-linked aggregates are too dense.

## 3. Types of Nanostructured Supports

The major nanostructured supports utilized for cellulase immobilization include nanoporous materials (MOFs [80], biochars [81,82,83], porous silica [84], etc.), nanohydrogels, polymer NPs, magnetic NPs, etc. The majority of these supports was employed for years for enzyme immobilization but in the last five years we can see innovations in the fabrication or modification of these nanomaterials to better adjust for cellulase loading and function.

### 3.1. Porous Nanomaterials

Porous materials with various pore sizes including those with hierarchical porosity have been explored for cellulase immobilization. Novel wrinkled mesoporous silica NPs possessing radial and hierarchical open pore structures have been developed and utilized for enzyme immobilization [27,48]. Varying (smaller, WSN, and larger, WSN-p) inter-wrinkle distances which, in turn, depended on the conditions of the silica NP preparation, the authors were able to successfully adsorb BG and cellulase most likely due to hydrogen bonding without damaging the enzyme secondary structures (Figure 5) [27]. Despite the absence of chemical bonds between enzymes and the support, the nanobiocatalysts displayed high stability probably due to a combination of high surface area and specific pores/folds, capturing enzymes.

A combination of mesoporous Fenton catalyst (Fe-MCM-48) and cellulase immobilized on the same support allows efficient depolymerization of chitosan [80]. This accomplishment demonstrates an innovation in the application of very dissimilar catalysts on the same support and in the same complex process.

A comparison of mesoporous silicas with 17.6 nm and 3.8 nm average pores demonstrated that the larger pores whose sizes are similar to a long axis of cellulase allows for higher enzyme loading [85]. On the other hand, 3.8 nm pores that are close in size to a short axis of cellulase provide a higher activity due to preservation of enzyme active sites. Thus, despite any possible preconceived notion that smaller pores could damage a secondary cellulase structure, authors’ thoughtful selection of pore sizes in two mesoporous materials allowed for a deeper understanding of restrictions or lack thereof in the choice of nanoporous supports.

### 3.2. Nanogels

In recent literature, there are only a few examples of nanogels, although they seem to possess a clear advantage: a swollen state of hydrogel can allow better access to immobilized cellulase, thus enhanced enzymatic activity. A nanogel based on poly(acrylic acid) prepared by inverse-phase microemulsion polymerization was utilized for cellulase adsorption [86]. It demonstrated high temperature tolerance, retaining 75% of activity at 80 °C as well as higher pH tolerance. A hybrid nanogel support where the PANI nanorods were formed in situ within nanogel prepared by electrospinning has been discussed in the section on the covalent attachment [72]. In this work, higher activity of the immobilized enzyme compared to the free enzyme was observed in the same temperature range, but the immobilized cellulase showed higher thermal and storage stability. Nanohydrogels were formed by grafting of carboxymethyl cellulose with acrylic polymers in the presence of GO sheets, whose role was to allow dual cross-linking via hydrogen bonding [87]. After cellulase encapsulation, the nanobiocatalyst was applied for the enhanced hydrolysis of lignocellulosic biomass and showed a remarkable increase in conversion of sugar beet pulp treated with alkaline. We think that an additional advantage of nanogels can be realized when they are pH or temperature responsive to allow for removal of the hydrolysis products that might be retained in the nanogel.

### 3.3. Polymer Particles

Polymer particles can be beneficial for a surface covalent attachment of enzymes if the polymers possess functional groups. Polymer NPs were prepared from a crosslinked copolymer of styrene and maleic anhydride using precipitation polymerization without a stabilizer followed by the covalent attachment of cellulase via anhydride groups [88]. Poly(styrene)-*b*-poly(styrene-*alt*-maleic anhydride) modified with nitrilotriacetic acid (NTA) self-assembled into micelles, whose modification with Ni^2+^ led to the attachment of *His*_6_-tagged cellulases (obtained from a bacterial cell) to produce core-shell NPs with cellulases in the outer layer (Figure 6) [89].

Such hierarchical structure allowed for cellulase exposure to the reaction media and preservation of cellulase conformation due to a soft support. A chitosan-cellulase nanohybrid has been prepared by self-assembly of chitosan in the presence of cellulase followed by immobilization on alginate beads [49].

### 3.4. Magnetic Nanostructured Supports

Magnetically responsive nanostructured supports are usually based on magnetic NPs. The use of magnetic NPs for the development of nanobiocatalysts has skyrocketed in recent years due to easy magnetic separation, allowing multiple reuses of nanobiocatalysts and making the processes more robust and economically and environmentally favorable. Magnetic NPs (most frequently iron oxide NPs) are normally functionalized to allow the enzyme attachment. To achieve that, such NPs are either coated with silica followed by the attachment of functional (amino) groups [65,67,71,90,91,92,93,94,95,96,97] or with a polymer containing reactive groups, for example, chitosan or other functional polymers [22,98,99,100,101,102,103,104,105,106,107,108]. The addition of metal ions (for example, copper) to amino-functionalized magnetic NPs allows for improved cellulase immobilization due to metal affinity [109]. For better protection of iron oxide NPs, Poorakbar et al. used a gold shell around magnetic NPs followed by the silica shell and functionalization with PEG and L-aspartic acid for a covalent cellulase attachment [110]. The other avenue for a magnetic nanobiocatalyst synthesis is realized when magnetic NPs are embedded into porous or polymer materials [111,112]. Even bare magnetite NPs have been utilized for adsorption of cellulase [113] or after functionalization with glutaraldehyde [114].

When Fe_3_O_4_ NPs coated with SiO_2_ were additionally functionalized with a copolymer shell consisting of poly(N-isopropylacrylamide-*co*-glycidyl methacrylate) P(NIPAM-GMA), more opportunities for nanobiocatalyst tuning were offered [115]. PGMA allows for a covalent attachment of cellulase, while PNIPAM is a temperature responsive polymer allowing control of swelling and deswelling with a temperature change. GO sheets modified with four-arm PEG macromolecules containing amino terminal groups were deposited on magnetic Fe_3_O_4_ NPs and employed for cellulase immobilization [116]. Similar functionalization was explored by the same group using solely magnetic NPs as support [117].

A fascinating example of the use of magnetic NPs was published by De Dios Andres et al. [118]. The authors employed a layer-by-layer technique to fabricate magnetic micromotors, whose upper layer was positively charged (Figure 7). After immobilization of cellulase, the micromotors were imbedded in paper chips to create diagnostic devices. Here, cellulase is needed to partially hydrolyze cellulose in the paper chips to increase the micromotor mobility. It is worth noting that an additional coating of micromotors with PEG diminishes their interaction with cellulose, allowing one to preserve chip integrity and to control the micromotor mobility. Although, no applications for these diagnostic devices were reported in the paper, it is a first example of using magnetic micromotors in the paper environment.

In the other example, to minimize the amount of Fe_3_O_4_ NPs used, the authors [119] adsorbed them on MWCNTs and utilized them as support for cellulase adsorption. Considering that MWCNTs are not a cheaper material than magnetite NPs, the whole idea of this construct seems questionable. On the other hand, naturally occurring halloysite nanotubes (clay materials) with attached iron oxide NPs and covalently bound cellulase (Figure 8) present a better alternative to MWCNT based nanomaterials [120]. The major advantage here is that the bulk of the nanobiocatalyst consists of cheap, naturally available material, making such a catalyst more promising for commercialization. In a similar approach, layering of magnetite NPs with double hydroxide nanosheets was utilized by Pei et al., leading to a magnetic support well suitable for covalent immobilization of cellulase via glutaraldehyde [121].

To create a beneficial support for cellulase immobilization, Papadopoulou et al. formed magnetic iron oxide NPs in hierarchical porous carbons containing macropores (>50 nm) as well as interconnected meso- and micropores [122]. The authors explored both covalent and non-covalent (adsorption) attachment of cellulase and determined that the covalent immobilization provides higher activity and stability upon reuse. Magnetic (Fe_3_O_4_) NPs coated by quaternized lignosulfonate and bearing pH-responsive properties were synthesized for immobilization and recovery of cellulase from biorefinery process waste [123]. Cellulase was immobilized or desorbed upon pH changes due to electrostatic interactions. Schnell et al. demonstrated a similar behavior based on electrostatic and coordination interactions on the surface of bare iron oxide NPs with carboxylic acid groups of cellulase [113]. The authors also discovered that the Fe^2+^:Fe^3+^ ratio influences the enzyme loading and activity.

An interesting magnetic support was proposed by Raza et al. [124]. For its fabrication, first hollow polymer particles were made by precipitation from bio-phenylpropene. This followed by the attachment of amino-functionalized Fe_3_O_4_ NPs with further modification by glutaraldehyde to form multi-layered magnetic hollow particles for a cellulase covalent attachment.

Magnetic core-shell MOFs for cellulase immobilization have been prepared via growing UIO-66-NH_2_ on the surface of poly(sodium 4-styrenesulfonate) modified Fe_3_O_4_ NPs [125]. This support allowed for high loading capacity of the enzyme and demonstrated high pH and thermal stability as well as better tolerance to formic acid and vanillin which are conventional inhibitors of the intermediates in lignocellulosic hydrolysis.

An original approach was proposed by Tan et al. to make a nanocomposite with oriented cellulase on chitosan/Fe_3_O_4_ NPs [126]. To accomplish that, the authors mixed cellulase with cellulose (a sacrificial template), tightly attaching the enzyme to the latter. Then, the mixture was embedded in chitosan, followed by the formation of magnetic NPs on chitosan periphery and hydrolysis of cellulose. This resulted in a hollow magnetically recoverable structure with cellulase in the stretched and most active conformation.

## 4. Attachment of Multiple Enzymes

Co-immobilization of two or more enzymes along with cellulase appears to be a winning strategy for the enhancement of saccharification of lignocellulosic biomass into bioethanol and other value-added chemicals. Several enzymes can be utilized to improve the immobilized cellulase performance. It is well known that laccase can oxidize lignin and phenolic compounds present in lignocellulosic biomass and thus prevents their negative influence on the cellulase biocatalytic activity [127,128]. Following this avenue, Kumar et al. co-immobilized laccase and cellulase on amino-functionalized magnetic NPs [129]. The nanobiocatalyst obtained allowed for a direct production of bioethanol from rice straw. The immobilization of laccase and cellulase on chiral mesoporous silica loaded with GO sheets was used for an environmental application—the degradation of methoxychlor in simulated polluted soils [130]. Here, the role of GO is to increase thermal and acid stability as well as reusability of the nanobiocatalyst. The attachment of lysozyme along with cellulase on magnetic NPs functionalized with amino groups allows for degradation of cell walls, thus, more efficient hydrolysis [23]. The covalent immobilization of cellulase and α-amylase (degrading starch) was carried out on amino-functionalized silica coated Fe_3_O_4_ NPs [93]. This magnetic nanobiocatalyst showed an excellent performance in extraction of anthocyanin from black rice and could be recommended for other food industry applications due to easy recovery and high efficiency.

A combination of covalently attached xylanase and cellulase improved cellulose degradation [62,70,131]. Xylanase allows for xylan hydrolysis [132] and is also employed for removal of the hemicellulose coating from cellulose microfibrils, leading to easier cellulose hydrolysis [133]. A covalent attachment of xylanase, cellulase, and amylase-cum-glucanotransferase on magnetic NPs yielded a multifunctional and magnetically recoverable catalyst for efficient saccharification of biomass [133]. Amylolytic glucanotransferase degrades starch [134], thus providing a more comprehensive hydrolysis of biomass waste.

A biocatalyst containing immobilized laccase, cellulase, and *β*-glucosidase has been studied in saccharification of four lignocellulosic biomass sources, such as *Typha angustifolia*, *Arundo donax*, *Saccharum arundinaceum*, and *Ipomoea carnea* [135]. The results showed that co-immobilization of three enzymes allows for an efficient, one-pot pretreatment for the bioethanol production. A cocktail of enzymes, such as cellulases, hemicellulases, chitinases, esterases, amylases, etc. found in holocellulase from *Aspergillus niger* SH3 was immobilized on several different NP classes and showed promise in hydrolysis of paddy straw pretreated with alkali [136]. The significant advantage of this approach is all enzymes are produced at the same time, do not require purification, and significantly simplify the nanobiocatalyst fabrication.

The same avenue was utilized by Muley et al. using fermentation broth of the *Aspergillus niger* culture containing cellulase, pectinase, and xylanase and covalently immobilizing them on magnetic NPs [26]. Here, pectinase degrades pectin, which improves saccharification of pectin-rich biomass [137]. A mixture of cellulase, xylanase, and *β*-1,3-glucanase (degrading glucan chitosan) were attached to silica coated iron oxide NPs containing amino groups on periphery and used for hydrolysis of sugar cane bagasse pulp into monomeric sugars with high activity and reusability (Figure 9) [138].

Four different enzymes, such as laccase, cellulase, *β*-galactosidase, and transglutaminase were either cross-linked forming CLEA or crosslinked in the presence of iron oxide NPs [139]. Magnetic CLEA allowed for easy catalyst separation, but in general, the authors did not demonstrate a remarkable enhancement of hydrolytic processes despite a combination of enzymes. This work reveals that lignocellulosic biomass hydrolysis can be inhibited by some other substances or factors, which the authors did not take into account.

Sometimes additional enzymes are immobilized along with cellulase to alter the final product of biomass waste hydrolysis. For example, the covalent attachment of cellulase and glucose oxidase on GO allowed for a direct transformation of cellulose to gluconic acid [28]. Concurrent covalent immobilization of four enzymes—cellulase, *β*-glucosidase, glucose oxidase, and horseradish peroxidase—on amino-functionalized magnetic NPs created a nanobiocatalyst for the cascade hydrolysis of cellulose to glucose [24].

A robust method for a covalent co-immobilization of enzymes has been developed by Dedisch et al. via adhesion-promoting peptides called Matter-tags [140]. The authors used three Matter-tags—*Cecropin A* (*CecA*), liquid chromatography peak I (LCI), and *Tachystatin* A2 (TA2)—that were connected to a green fluorescent protein and two enzymes, phytase and cellulase. It was discovered that LCI is a universal adhesion promoter, which allowed for immobilization of both enzymes on various polymers, metals, and silicon wafer within ~10 min at ambient temperature. The authors believe any enzymes can be immobilized in this way, making it a universal platform for the multiple enzyme immobilization.

## 5. Strategies to Improve Interactions between Immobilized Cellulase and Cellulosic Biomass

It is noteworthy that one of the problems of catalyzing the hydrolysis of cellulosic biomass with immobilized cellulase is interactions of two solids. To counteract this shortcoming, an interesting strategy for the development of the support was utilized by modification of a copolymer of methacrylic acid and methyl methacrylate which is pH-responsive and allows a phase transition between soluble and insoluble forms [141]. A functionalization of this polymer with iminodiacetic acid (IDA) and Ni^2+^ ions created immobilization points for *His*-tagged endoglucanase EG5C-1 (cellulase) due to its affinity interaction with Ni^2+^ (Figure 10). The ability of this biocatalyst to be reversibly transformed between soluble/insoluble phases allowed for better interactions with cellulose at high pH and separation in an insoluble form when pH is decreased after the reaction.

To allow a switchable temperature–pH dual-responsive material, Zhu et al. synthesized a random copolymer from single-strand DNA-functionalized acrylamide monomer N-isopropylacrylamide, and N-isopropylmethylacrylamide [142]. Cellulase was physically encapsulated in the hydrogel at pH 5.4, which could be further released at pH 7.4 when the hydrogel becomes unstable and hydrolyzes cellulose.

## 6. Conclusions

Nanobiocatalysts based on cellulase immobilized on nanostructured supports have been utilized mainly for catalytic hydrolysis of biomass waste as well as in food processing and environmental applications. An analysis of the latest trends presented in this review demonstrates that there were impressive innovations in the immobilization methods and the support structures in the last five years. One of the most striking examples includes a site-specific enzyme immobilization on a stimuli responsive microgel via sortase-mediated enzyme immobilization, thus, making immobilization more targeted and improving the interaction with cellulose due to stimuli responsive support. The other example demonstrates that an unusual structure of the support—wrinkled mesoporous silica NPs—allows for an efficient a stable nanobiocatalysts obtained via a simple cellulase adsorption due to the unique character and morphology of the support. Adsorbed cellulase was significantly stabilized and activated due to modification of the support with charge changing macromolecules or those altering the hydrophobicity-hydrophilicity balance. Support free crosslinked enzyme nanoflowers allowed for the authors to achieve a similar goal due to fluffy enzymatic structures with good affinity to cellulosic biomass.

We believe the co-immobilization of multiple enzymes on various supports is the most promising avenue for the future nanobiocatalyst development. It could significantly improve the outcome of cellulose saccharification to sugars or bioethanol due to efficient hydrolysis of lignin, starch, etc., which accompany cellulose in biomass waste and are detrimental for cellulose hydrolysis. Multiple enzyme immobilization also allows one to obtain completely different products from cellulosic biomass due to specific enzymes. Additionally, the utilization of magnetically separable supports makes the biocatalyst preparation more robust and facilitates biocatalytic processes due to magnetic nanobiocatalyst recovery. Finally, the pathways to optimize the contact between the immobilized cellulase and cellulosic biomass via stimuli responsive materials appear favorable for further development of nanobiocatalysts for cellulosic biomass processing.

## Data Availability

Not applicable.

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
