# Peer review of "Cellulase Immobilization on Nanostructured Supports for Biomass Waste Processing"

_nanomaterials, 2022, doi:10.3390/nano12213796_

Round 1

Reviewer 1 Report

This work summarized the recent developments in the immobilization of cellulase on nanostructured supports, and introduced methods of cellulase immobilization, types of nanostructured supports, strategies to attach multiple enzymes, and improve interactions between immobilized cellulase and cellulosic biomass. The manuscript has rich content and clear logic, it is informative for deepening understanding of cellulase Immobilization on nanostructured Supports.

Several specific comments are listed as follows:

1.      It is recommended that a schematic illustration Figure is provided as Figure 1 to introduce the main content of the review.

2.      In section 2.2, a brief introduction of the encapsulation method should be provided before providing e specific examples.

3.      The advantages of disadvantages of the cellulase immobilization method should be included to provide a more comprehensive understanding of the features of these approaches.

4.      As the review entitled “Cellulase Immobilization on Nanostructured Supports for Biomass Waste Processing”, a brief summary regarding the applications of cellulase immobilized nanostructured support in biomass waste processing should be added in the last part of the manuscript.

Other comments

1.      There’re some typo errors. For example, in Line 16, 24, 57, 79, “cellulase” was misspelled as “celullase”.

2.      Figure 2 didn’t include the reference number and citing permission.

3.      Some pictures are not clear, such as Figure 9, Figure 10.

Author Response

Comment 1:     It is recommended that a schematic illustration Figure is provided as Figure 1 to introduce the main content of the review.

Response 1: Scheme 1 has been added in the Introduction to illustrate the main content of the review.

Comment 2:      In section 2.2, a brief introduction of the encapsulation method should be provided before providing e specific examples.

Response 2: The brief introduction in section 2.2 has been added.

Comment 3:      The advantages of disadvantages of the cellulase immobilization method should be included to provide a more comprehensive understanding of the features of these approaches.

Response 3: This information has been added in sections 3.1-3.4.

Comment 4:      As the review entitled “Cellulase Immobilization on Nanostructured Supports for Biomass Waste Processing”, a brief summary regarding the applications of cellulase immobilized nanostructured support in biomass waste processing should be added in the last part of the manuscript.

Response 4: A brief summary of applications of cellulase based nanobiocatalysts has been added in Conclusion.

Other comments

Comment 1a.      There’re some typo errors. For example, in Line 16, 24, 57, 79, “cellulase” was misspelled as “celullase”.

Response 1a: Typos were corrected.

Comment 2a:     Figure 2 didn’t include the reference number and citing permission.

Response 2a: This Figure was prepared by us. We did not reproduce the manuscript figure. That is why no reference number or citing permission is necessary.

Comment 3a:      Some pictures are not clear, such as Figure 9, Figure 10.

Response 3a: All figures were uploaded from the corresponding journal websites and copied at the highest available resolution.

Reviewer 2 Report

This manuscript is a comprehensive review of cellulase immobilization and attachment to nanostructured supports. The article is very interesting and well-written.

Author Response

We are thankful to the reviewer of the favorable review. 

Reviewer 3 Report

The review written by Sulman et al. presents a good summary of the recent literature on the immobilization of enzymes involved in the depolymerization of cellulose from biomass waste. I have found it interesting and easy to read. After briefly reviewing the main methods available for the immobilization of enzymes, using specific examples of cellulases, they describe different supports and the very interesting innovations described in different works to improve the activity of cellulases. Likewise, they collect some results of co-immobilization of cellulases with enzymes that act on other components of lignocellulose. They end up with new materials capable of responding to external stimuli, keeping the enzyme soluble during hydrolysis to improve the enzyme-substrate interaction, and capturing it once the reaction is complete.

I only want to point a few things.

- In lines 50-51 it is said that “This growing interest is explained by a possibility of nanobiocatalysts to overcome deficiencies of traditional biocatalysts (immobilized enzymes)”. This sentence is not clear to me, I think that nanobiocatalysts overcome deficiencies of traditional immobilization carriers, not of traditional biocatalysts.

In section 3.2. nanogels are said to have a clear advantage, which is the better accessibility of the substrate in its hydrated state (lines 263-265). However, the last line undermines their usefulness because of the possibility of retaining the reaction product. I do not know if this is deduced from the results described in references 86, 87 and 72 cited in this epigraph or if this is a hypothesis of the authors. In any case, I consider it interesting to comment on the results of these studies in more detail and eliminate hypotheses if they are not supported by data.

- Likewise, I find the work of reference 118 (line 319) very attractive, but the application of the invention is insufficiently explained. Is it usable to process wastes? What kind of diagnostic devices are based on it?

- The following paragraph (line 332) seems to be at first a continuation of the previous one, but further reading shows that this is not the case. Please rewrite the first sentence of said paragraph to avoid this confusion for the reader.

Author Response

Comment 1: In lines 50-51 it is said that “This growing interest is explained by a possibility of nanobiocatalysts to overcome deficiencies of traditional biocatalysts (immobilized enzymes)”. This sentence is not clear to me, I think that nanobiocatalysts overcome deficiencies of traditional immobilization carriers, not of traditional biocatalysts.

Response 1: We agree with the reviewer and made changes in the manuscript.

Comment 2: In section 3.2. nanogels are said to have a clear advantage, which is the better accessibility of the substrate in its hydrated state (lines 263-265). However, the last line undermines their usefulness because of the possibility of retaining the reaction product. I do not know if this is deduced from the results described in references 86, 87 and 72 cited in this epigraph or if this is a hypothesis of the authors. In any case, I consider it interesting to comment on the results of these studies in more detail and eliminate hypotheses if they are not supported by data.

Response 2: Indeed, this was our hypothesis based on the understanding of such nanocarriers. In the revised manuscript we added more details of the mentioned studies and revised our opinion. Considering that this is a critical review, we feel it is important to express our take on the matter.

Comment 3: Likewise, I find the work of reference 118 (line 319) very attractive, but the application of the invention is insufficiently explained. Is it usable to process wastes? What kind of diagnostic devices are based on it?

Response 3: In the revised manuscript we added more details from reference 118. Please note that in this work cellulose is from the paper  and it is not processed, thus processing the waste is not relevant. Partial cellulose (paper) hydrolysis with cellulase is used to afford mobility of micromotors.

Comment 4: The following paragraph (line 332) seems to be at first a continuation of the previous one, but further reading shows that this is not the case. Please rewrite the first sentence of said paragraph to avoid this confusion for the reader.

Response 4: The first sentence of the said paragraph was revised according to the reviewer advice.